# Superconductivity emerging from a stripe charge order in IrTe₂ nanoflakes

Sungyu Park[1,14], So Young Kim[1,2,14], Hyoung Kug Kim[2], Min Jeong Kim[1,3], Taeho Kim[1,3], Hoon Kim[1,2], Gyu Seung Choi[1,2], C. J. Won [4,5], Sooran Kim [6], Kyoo Kim [7], Evgeny F. Talantsev [8,9], Kenji Watanabe [10], Takashi Taniguchi [11], Sang-Wook Cheong [4,5,12], B. J. Kim [1,2], H. W. Yeom [1,2], Jonghwan Kim[1,2,3 ✉], Tae-Hwan Kim [2,5,13 ✉] & Jun Sung Kim [1,2 ✉]

Superconductivity in the vicinity of a competing electronic order often manifests itself with a superconducting dome, centered at a presumed quantum critical point in the phase diagram. This common feature, found in many unconventional superconductors, has supported a prevalent scenario in which fluctuations or partial melting of a parent order are essential for inducing or enhancing superconductivity. Here we present a contrary example, found in IrTe₂ nanoflakes of which the superconducting dome is identified well inside the parent stripe charge ordering phase in the thickness-dependent phase diagram. The coexisting stripe charge order in IrTe₂ nanoflakes significantly increases the out-of-plane coherence length and the coupling strength of superconductivity, in contrast to the doped bulk IrTe₂. These findings clarify that the inherent instabilities of the parent stripe phase are sufficient to induce superconductivity in IrTe₂ without its complete or partial melting. Our study highlights the thickness control as an effective means to unveil intrinsic phase diagrams of correlated van der Waals materials.

[1] Center for Artificial Low Dimensional Electronic Systems, Institute for Basic Science, Pohang, Korea. [2] Department of Physics, Pohang University of Science and Technology, Pohang, Korea. [3] Department of Materials Science and Engineering, Pohang University of Science and Technology, Pohang, Korea. [4] Laboratory for Pohang Emergent Materials, Pohang Accelerator Laboratory, Pohang, Korea. [5] Max Planck POSTECH/Korea Research Initiative, Pohang, Korea. [6] Department of Physics Education, Kyungpook National University, Daegu, Korea. [7] Korea Atomic Energy Research Institute (KAERI), Yuseong-gu, Daejeon, Korea. [8] M.N. Mikheev Institute of Metal Physics, Ural Branch, Russian Academy of Sciences, Ekaterinburg, Russia. [9] NANOTECH Centre, Ural Federal University, Ekaterinburg, Russia. [10] Research Center for Functional Materials, National Institute for Materials Science, Tsukuba, Japan. [11] International Center for Materials Nanoarchitectonics, National Institute for Materials Science, Tsukuba, Japan. [12] Rutgers Center for Emergent Materials and Department of Physics and Astronomy, Rutgers University, Piscataway, NJ, USA. [13] Asia Pacific Center for Theoretical Physics (APCTP), Pohang, Korea. [14]These authors contributed equally: Sungyu Park, So Young Kim. ✉email: jonghwankim@postech.ac.kr; taehwan@postech.ac.kr; js.kim@postech.ac.kr

Transition metal dichalcogenides (TMDCs) provide a prototypical quasi-two-dimensional system, possessing various electronic instabilities to periodic charge modulations[1]. These instabilities often induce complex charge-density-wave (CDW) phases with different commensurability conditions[2–5], sometimes associated with Mott-[2] or excitonic insulating[6] phases. Upon chemical doping[4,5,7,8], these phases are commonly driven into a superconducting phase, resulting in a characteristic dome-shaped phase diagram, reminiscent of those found in other unconventional superconductors[9–11] (Fig. 1a). Understanding the complex interplay between charge ordering and superconducting instabilities in TMDCs is a long standing issue, which has often been hampered by presence of quenched disorders, introduced in chemical doping. Recently, utilising the weak van der Waals (vdW) coupling between layers, TMDCs were found to be thinned down to atomic length scale[12], comparable with the coherence lengths of their various electronic orders. This offers another effective way to tune stability or properties of the competing phases, as demonstrated for $1T$-$TaS_2$[13–15] and $NbSe_2$[16,17], in which distinct thickness dependence of the transition temperatures is observed for superconducting and CDW phases.

$IrTe_2$ is one of the TMDCs in vdW structure (Fig. 1b), showing a similar dome-shaped superconducting phase diagram[18–23]. $IrTe_2$ undergoes a stripe charge ordering transition at $T_s \sim 260$ K, and by suppressing it with e.g. chemical doping[18–23] the superconducting phase eventually appears, similar to TMDCs hosting the parent CDW orders[4,5,7,8]. The stripe order in $IrTe_2$, however, is accompanied by the first-order structural transition involving in-plane Ir–Ir dimerization and interlayer Te–Te depolymerisation[24,25], which forms stripe patterns with a predominant period of $5a_0$ ($a_0$, the $a$ axis lattice constant) as depicted in Fig. 1c and d[19,26]. No clear evidence of gap opening in the Fermi surface (FS) is observed[27,28], unlike the typical CDW gap formation in TMDCs[29]. Instead, FS reconstruction to the so-called cross-layer two-dimensional (2D) state[30,31] occurs due to suppression of the density of states (DOS) in the planes of Ir–Ir dimers running across the vdW gaps. These aspects suggest that the relationship between the stripe and superconducting orders in $IrTe_2$ may differ significantly from those of other TMDCs, as indicated by recent discoveries on the superconductivity in quenched or thinned $IrTe_2$ crystals[32–34]. Here using Raman spectroscopy, scanning tunnelling microscopy, and transport property measurements, we found that the parent stripe phase encompasses the whole superconducting dome in the thickness-dependent phase diagram (Fig. 1a). This unusual coexistence of the stripe and superconducting orders significantly increases the interlayer coherence length and the coupling strength of superconductivity in $IrTe_2$ nanoflakes, revealing the collaborative role of the stripe order to the superconductivity in $IrTe_2$.

## Results

**Thickness dependent phase transitions.** In order to vary the thickness of $IrTe_2$, we employed the mechanical exfoliation method of single crystals and obtained thin flakes with thickness ($d$) down to ~10 nm, which show a systematic thickness dependence of the in-plane resistivity $\rho$ (Fig. 1e). For the temperature sweeps in both directions, we took a slow cooling rate of ~0.5 K/min, in order to minimise inhomogeneous domain formation of the stripe-charge-ordered and charge-disordered phases, found in rapid-cooled $IrTe_2$ crystals[32,33,35]. The temperature dependent $\rho(T)$ follows a metallic temperature dependence with abrupt changes at $T_{s,dn}$ and $T_{s,up}$, due to the first-order stripe ordering transition as found in bulk crystals[24,25]. These resistive anomalies across the transitions, however, become much smaller in size with reducing $d$, and eventually disappears for $d < 50$ nm. This does not mean full suppression of

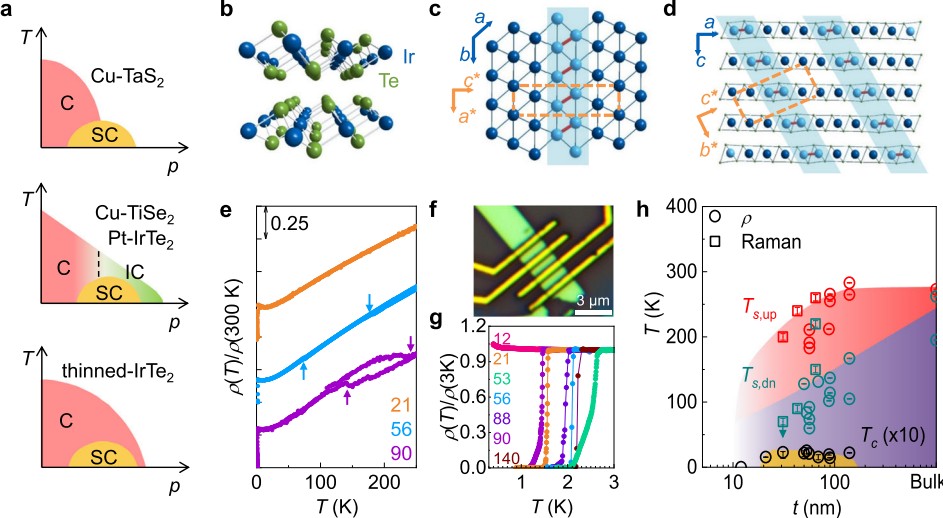

**Fig. 1 Structure and phase diagram of $IrTe_2$ nanoflakes. a** Schematic phase diagrams of TMDCs as a function of control parameter $p$, showing commensurate (C), incommensurate charge order (IC), and superconductivity (SC). Three different types of dome-shaped superconducting phase diagram, where the dome lies at the centre of a presumed quantum critical point (top), near the C-IC transition line (middle), or well inside the parent order (bottom). **b** Crystal structure of $IrTe_2$. **c, d** Schematic illustrations of the stripe order in $IrTe_2$ below $T_s$. The Ir-Ir dimerization (red) with a modulation vector $\mathbf{q} = (\frac{1}{5}, 0, \frac{1}{5})$ is depicted (blue shade) on the triangular Ir layer (**c**) and across the stacked layers (**d**). The crystallographic axes for the high-$T$ ($a$, $b$, and $c$) and the low-$T$ ($a^*$, $b^*$, and $c^*$) structures are shown, together with the unit cell of the stripe phase (orange box). **e** Temperature dependence of the normalised resistivity $\rho(T)/\rho(300$ K$)$ for $IrTe_2$ crystals with thickness ($d$) of 21, 56, and 90 nm. For clarity, $\rho(T)/\rho(300$ K$)$ curves are offset vertically. Transition temperatures $T_{s,up}$ and $T_{s,dn}$ are determined (arrows) in opposite temperature sweeps. **f** Optical microscope image of a 56-nm-thick $IrTe_2$ crystal. **g** $\rho(T)/\rho(3$ K$)$ curves for $IrTe_2$ crystals with $d = 12$–$140$ nm. **h** Phase diagram of $IrTe_2$ nanoflakes as a function of thickness $d$, obtained by transport (circle) and Raman spectroscopy (square) measurements. The transition temperatures $T_{s,up}$ (red) and $T_{s,dn}$ (blue) during warming and cooling are plotted with the superconducting transition temperature $T_c$ (black), scaled by a factor of 10 for clarity. The error bars from the resistivity and Raman spectroscopy are defined by the width of the corresponding resistive transitions and the temperature step of 5 K between the measurements, respectively.

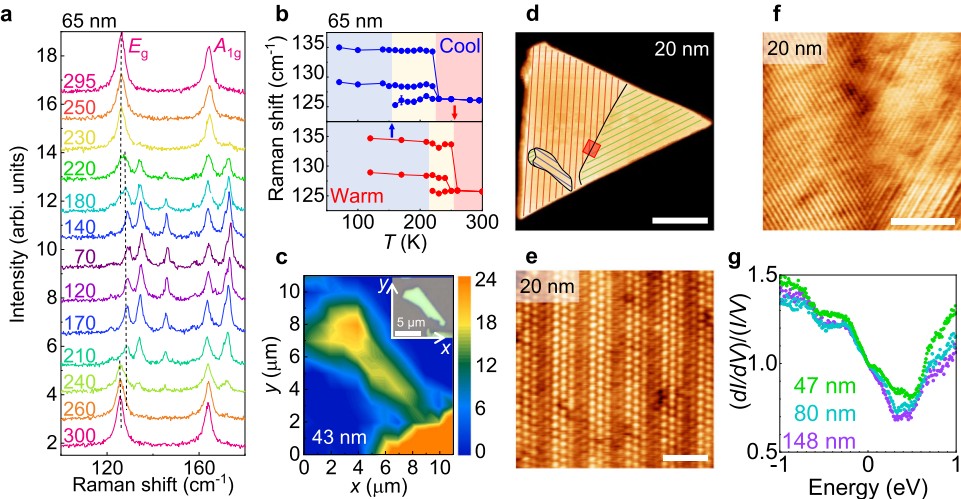

**Fig. 2 Stripe charge ordering formation in IrTe₂ nanoflakes. a, b** Raman spectra (**a**) and corresponding temperature dependent Raman frequency (**b**) of a 65-nm-thick IrTe$_2$ nanoflake at various temperatures during cool-down and warm-up procedures. At $T < T_s$, the Raman modes, $E_g$ at 126 cm$^{-1}$ and $A_{1g}$ at 165 cm$^{-1}$, split into multiple peaks as the flake forms the stripe charge order in **a**. The temperature ranges for the normal (red), the stripe (blue), and intermediate coexistence (yellow) phases, are identified in **b**, during cooling (upper panel) and warming (lower panel). Transition temperatures $T_{s,dn}$ and $T_{s,up}$ are indicated by the arrows, respectively. **c** Spatial profile of Raman intensity map for 129 cm$^{-1}$ for a 43-nm-thick IrTe$_2$ nanoflake at $T \sim 70$ K. Inset: optical microscope image of the flake. **d** Large-scale scanning tunnelling microscopy (STM) image of a 20-nm-thick IrTe$_2$ flake at $T = 85$ K (scale bar, 300 nm). The flake has only stripe-phase charge-ordered domains, illustrated by red, green, and blue lines. Black lines indicate domain boundaries between three equivalent stripe-phase charge-ordered domains. **e** Atomically resolved STM image of **d** at $T = 85$ K representing a uniform striped area with 5 × 1 surface reconstruction (scale bar, 2 nm). **f** Zoomed-in STM image of two charge-ordered phases indicated by red square in **d** showing that the two phases coexist at the boundary (scale bar, 20 nm). **g** Scanning tunnelling spectroscopy (STS) spectra at $T = 85$ K taken on IrTe$_2$ nanoflakes with $d = 47$, 80, and 148 nm, as indicated in the plot.

the stripe order in thin nanoflakes, since the size of the resistive anomaly is known to be strongly suppressed by introducing strain, reducing the sample volume, and increasing the cooling rate even in bulk samples[34]. Rather, the stripe order is found to be stable in all the nanoflakes we studied, as discussed below (Fig. 2). At low temperatures, all the nanoflakes, except the thinnest one with $d = 12$ nm, exhibit a superconducting transition as found in the temperature-dependent normalised resistance $\rho(T)/\rho(3$ K) (Fig. 1g). The superconducting transition temperature, defined as a 50% resistive transition, is $T_c = 1.43$–2.64 K, somewhat lower than $T_c \sim 3$ K for the optimally doped bulk IrTe$_2$[18,19], mostly due to the 2D nature as discussed below (Fig. 3). Unlike the stripe charge ordering transition, the superconductivity is found to be quite stable in nanoflakes. The superconducting transition temperatures and widths remain almost the same in different thermal cycling (Supplementary Fig. 7).

The stripe ordering transition of IrTe$_2$ nanoflakes is characterised by Raman spectroscopy. For bulk IrTe$_2$, two Raman active modes, $E_g$ at 126 cm$^{-1}$ and $A_{1g}$ at 165 cm$^{-1}$, in a trigonal structure (space group $P\bar{3}m1$) split into multiple peaks due to the lowered symmetry and the emergence of a super unit cell for the stripe order below $T_s$[30,36]. Raman spectra taken from the 65-nm-thick nanoflake with sequential decrease and increase of temperature (Fig. 2a) reveal that both $E_g$ and $A_{1g}$ Raman modes at room temperature split into multiple Raman modes at $T = 70$ K, well below $T_s$, consistent with previous studies on the bulk[36]. This behaviour is also shown for the flakes with different thicknesses (Supplementary Fig. 2a and Fig. 3a) and confirms formation of the stripe charge order in our IrTe$_2$ nanoflakes.

In IrTe$_2$ nanoflakes, however, the temperature dependence of the stripe phase evolution is distinct from the bulk case. We found that below $T_s$ there is an intermediate temperature range (yellow range in Fig. 2b) where the high temperature Raman modes coexist with the low temperature modes, for both temperature sweeps. This contrasts to the abrupt change found

in bulk IrTe$_2$ with a negligible coexistence range[36] and indicates macroscopic phase separation of the normal and stripe phases in IrTe$_2$ nanoflakes at the intermediate temperature. We defined $T_{s,dn}$ and $T_{s,up}$ as the temperatures where the contribution from the normal and stripe phase disappears during the cool-down and warm-up procedures, respectively (Fig. 2b). As the flake thickness decreases, the transition temperatures monotonically decrease, which are in good agreement with those from $\rho(T)$ (Fig. 1h). In addition, we measured the Raman spectroscopy on IrTe$_2$ flakes, cooled down to $T = 4$ K, just above $T_c \sim 2$ K. For IrTe$_2$ flakes with various thicknesses from 10 nm to 174 nm, which cover the whole superconducting dome in the thickness-dependent phase diagram, we observed clear splitting of Raman modes, consistent with the stripe-charge-order formation (Supplementary Fig. 3). These results clearly show that the region of the stripe phase overlaps with the entire superconducting dome in a wide range of thickness.

**Coexistence of stripe order and superconductivity.** This phase diagram significantly differs from the doping-dependent phase diagram of bulk IrTe$_2$[18–23]. In the bulk case, the stripe and superconducting phases are mutually exclusive, and the coexisting region appears in the very narrow doping range[18–23]. Even in this coexisting phase, two ordered phases are macroscopically separated[22,23]. Such a macroscopic phase separation is also suggested in the super-cooled case[32,33,35]. However, in IrTe$_2$ nanoflakes, the stripe phase completely covers the whole regions of the sample, as confirmed by the spatial mapping of Raman signal, taken at ~ 70 K well below $T_s$ (Fig. 2c). The intensity map of the 129 cm$^{-1}$ Raman mode, which is the hallmark of the stripe phase, reveals a strong Raman intensity profile over the entire nanoflake as in the optical microscope image (Fig. 2c inset), while the signal from the normal phase is absent (Supplementary Fig. 2c). This indicates that the stripe phase dominantly prevails in the

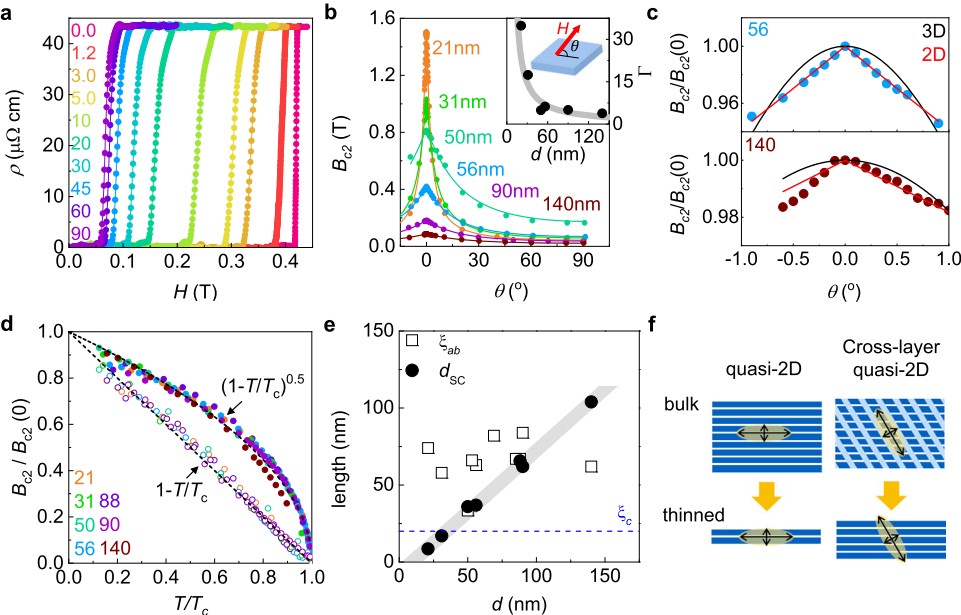

**Fig. 3 Two dimensional superconductivity of IrTe$_2$ nanoflakes. a**, Magnetic field dependence of $\rho(H)$ of a 56-nm-thick IrTe$_2$ nanoflake, measured with different field orientations $\theta$ at $T = 0.35$ K. **b**, Upper critical field $B_{c2}$ as a function of field angle $\theta$ for IrTe$_2$ nanoflakes with different thickness ($d$) at $T = 0.35$ K, together with the fit (solid line) to the 2D Tinkham model. Inset: the anisotropy factor $\Gamma = B_{c2}^{ab}/B_{c2}^{c}$ as a function of $d$, following $1/d$ dependence (grey line). Schematic illustration shows the field orientation $\theta$. **c**, Angle dependence of $B_{c2}(\theta)$ of IrTe$_2$ nanoflakes with $d = 56$ and 140 nm at $T = 0.35$ K. Good agreement with the 2D Tinkham model (red), rather than the 3D Ginzburg-Landau model (black), confirms the 2D superconductivity. **d**, Normalised $B_{c2}/B_{c2}(0)$ as a function of $T/T_c$ for IrTe$_2$ nanoflakes with different $d$. All data collapse into dashed lines described by $1 - T/T_c$ and $(1 - T/T_c)^{1/2}$ for $B\|c$ (open circles) and $B\|ab$ (solid circles), respectively. **e**, Ginzburg-Landau coherence length $\xi_{ab}$ (square) and the effective superconducting thickness $d_{SC}$ (circle) as a function of $d$. $\xi_{ab}$ is nearly independent of $d$, whereas $d_{SC}$ grows linearly with $d$ ($d_{SC} \sim 0.8d$) and exceeds $\xi_c$ of doped bulk IrTe$_2$. **f**, Schematic illustration of the size effect of vdW superconductors. In normal vdW superconductors with a large anisotropy $\xi_c \ll \xi_{ab}$, 2D superconductivity appears only for a-few-layer-thick crystals. In IrTe$_2$ with a stripe order and the resulting cross-layer quasi-2D state, the increased $\xi_c \sim \xi_{ab}$ induces 2D superconductivity in relatively thick nanoflakes.

macroscopic length scale and serves as a normal state for the superconductivity, in contrast to the doped bulk IrTe$_2$ case.

The dominant stripe phase formation is further confirmed by scanning tunnelling microscopy (STM) for a representative nanoflake with $d = 20$ nm (Fig. 2d). At room temperature, the hexagonal lattice of top-most Te atoms is clearly resolved by STM in all IrTe$_2$ nanoflakes (Supplementary Fig. 4b). When cooled down below the stripe ordering temperature ($T_{STM} = 85$ K $< T_s$), the ultrathin nanoflake develops clear stripe patterns with a period of $5a_0$ due to the charge ordering and dimerization of Ir atoms (Fig. 2e), as observed in bulk crystals[25,26]. By scanning over the nanoflake with sufficient spatial resolution (Fig. 2f), we confirm that the whole surface of the flake hosts one or two predominant stripe phases among three energetically equivalent phases (Fig. 2d), and the stripe patterns are often oriented nearly parallel to the long edges of nanoflakes. In thicker nanoflakes (Supplementary Fig. 5a), stripe domain patterns became more complex, similar to bulk IrTe$_2$[35]. We note that even with fast cooling at ~ 1 K/sec, neither thin nor thick nanoflakes show the hexagonal phase that is often observed in either super-cooled or doped bulk IrTe$_2$[22,35] and considered to be responsible for the superconductivity. Consistently, in the scanning tunnelling spectroscopy (STS) measurements on IrTe$_2$ nanoflakes, we observe similar local DOS over the wide range of thicknesses. In STS spectra (Fig. 2g), obtained on different IrTe$_2$ nanoflakes with $47 \leq d \leq 148$ nm, the local spectral features are qualitatively consistent with the total DOS for the stripe phase of bulk IrTe$_2$, as estimated using first principle calculations[30]. Our results from STM and Raman spectroscopy provide strong evidence that the

superconductivity emerges from the preexisting stripe phase in IrTe$_2$ nanoflakes.

**2D superconductivity**. Now we focus on the effect of the underlying stripe order on the superconducting properties. To address this issue, we investigated the upper critical field $B_{c2}$ of each nanoflake as a function of field orientation below $T_c$. Figure 3a shows the resistivity $\rho(H)$ curves of a representative nanoflake with $d = 56$ nm, collected at $T = 0.35$ K under a magnetic field, for which the angle $\theta$ is defined with respect to the $ab$ plane. The anisotropy of $B_{c2}$, $\Gamma = B_{c2}^{ab}/B_{c2}^{c} \sim 5.3$ for $d = 56$ nm, becomes stronger with lowering $d$ and reaches up to $\Gamma \sim 38$ for $d = 21$ nm (Fig. 3b), an order of magnitude larger than $\Gamma \sim 2$ of doped bulk IrTe$_2$[37]. This large increase in $\Gamma$ with moderate changes in $T_c$ (Fig. 1g) and the in-plane coherence length $\xi_{ab}(0)$ (Fig. 3e) can only be explained by 2D superconductivity. In the Tinkham model of 2D superconductivity[38], the angle dependent $B_{c2}(\theta)$ at the zero-temperature limit is described by $|B_{c2}(\theta) \sin \theta/B_{c2}^{c}| + (B_{c2}(\theta) \cos \theta/B_{c2}^{ab})^2 = 1$, where $B_{c2}^{ab} = (\sqrt{12}\Phi_0)/(2\pi\xi_{ab}(0)d_{SC})$ and $B_{c2}^{c} = \Phi_0/(2\pi\xi_{ab}(0)^2)$ ($\Phi_0$, a flux quantum). Thus by reducing the effective thickness of the superconducting layer $d_{SC}$, $\Gamma$ becomes large with a constant $\xi_{ab}(0)$, and a discontinuous cusp in the $B_{c2}(\theta)$ curve near $\theta = 0°$ is expected. These predictions are distinct from those of the Ginzburg-Landau model for anisotropic three-dimensional (3D) superconductors[38], as described by $B_{c2}(\theta) = B_{c2}(0°)/\sqrt{\Gamma^2\sin^2\theta + \cos^2\theta}$. All $B_{c2}(\theta)$ curves exhibit a clear cusp near $\theta = 0°$ and are successfully fitted by the 2D Tinkham model rather than the anisotropic 3D model (Fig. 3b and Supplementary

Fig. 9). For $d = 140$ nm, the thickest sample, $B_{c2}(\theta)$ slightly deviates from the 2D model, but still far from 3D model (Fig. 3c). These results demonstrate that IrTe$_2$ nanoflakes with $d \lesssim 140$ nm clearly show the 2D superconductivity.

The temperature dependence of $B_{c2}(T)$ under in-plane ($B \| ab$) and out-of-plane ($B \| c$) magnetic fields further confirms the 2D superconductivity of IrTe$_2$ nanoflakes. For seven samples with different $d$'s, we determined $B_{c2}(T)$ by taking 50% of the resistive transition as a function of the normalised temperature $t = T/T_c$ (Fig. 3d and Supplementary Fig. 8). The out-of-plane $B_{c2}^c(t)$ is almost the same, following the linear dependence (Fig. 3d), as observed in the doped bulk sample[37]. The in-plane $B_{c2}^{ab}(t)$ increases strongly with lowering $d$, but the normalised $B_{c2}(t)/B_{c2}(0)$ curves for all samples collapse into a single curve following the 2D Ginzburg-Landau model[38], $B_{c2}^{ab}(t) = \frac{\Phi_0}{2\pi} \frac{\sqrt{12}}{\xi_{ab}^c d_{SC}} (1 - t)^{1/2}$. Using $\xi_{ab}(0)$, estimated from the observed $B_{c2}^c(0)$, we obtained $d_{SC}$ is $\sim 80\%$ of the measured thickness $d$ (Fig. 3e). Considering that 2D superconductivity is induced at $d$ ($\sim d_{SC}$) smaller than the out-of-plane coherence length $\xi_c$, i.e. $d < \xi_c$, we conclude that $\xi_c(0)$ of IrTe$_2$ nanoflakes should be larger than the maximum value of $d_{SC} \sim 100$ nm, obtained in experiments (Fig. 3e), This value is significantly larger than the typical $\xi_c(0) \sim 25$ nm of doped IrTe$_2$ bulk samples[37], and comparable to the in-plane coherence length $\xi_{ab}(0) \sim 70$ nm[39] (Fig. 3e). These findings clearly indicate that the superconducting characteristics of IrTe$_2$ nanoflakes are distinct from those of the doped IrTe$_2$.

**Characteristics of superconductivity coexisting stripe order.** The drastically increased $\xi_c(0)$ in IrTe$_2$ nanoflakes is a consequence of the coexisting stripe order. In anisotropic superconductors, the interlayer coherence length $\xi_c(0)$ is determined by the superconducting gap ($\Delta_{SC}$) and the Fermi velocity ($v_F^c$), i.e. $\xi_c(0) \propto v_F^c / \Delta_{SC}$. Assuming a similar $\Delta_{SC}$, $\xi_c(0)/\xi_{ab}(0) \approx v_F^c/v_F^{ab} \gtrsim 1$ seems incompatible with the vdW structure of IrTe$_2$. This however can be explained by considering the coexisting stripe order. In the stripe phase of IrTe$_2$, Ir-Ir dimerization and Te-Te depolymerisation produce conducting planes between the dimer planes, running across the vdW gaps (Fig. 1d). This cross-layer 2D conducting state[30] affects the electronic structure such that the interlayer $v_F^c$ is even larger than the in-plane $v_F^{ab}$, which greatly increases $\xi_c(0)$ (Fig. 3f). Unlike the conventional vdW superconductors in which 2D superconductivity can only be induced in a-few-layer-thick crystals, IrTe$_2$ hosts 2D superconductivity in the relatively thick crystals due to the microscopically coexisting stripe order.

The distinct superconducting nature in IrTe$_2$ nanoflakes is also found in their superconducting gap as compared to the doped bulk. Figure 4a presents the current-voltage (IV) characteristics at different temperatures for a representative nanoflake with $d = 21$ nm. Near $T_c \approx T_{BKT}$, they follow Berezinskii-Kosterlitz-Thouless transition for 2D superconductivity (Fig. 4b), in which the exponent $\alpha$ extracted from $V \propto I^\alpha$ crosses $\alpha = 3$ at $T_{BKT}$. Well below $T_c$, the self-field critical current density $J_{c,sf}(T)$ can be obtained from the IV characteristics with variation of temperature, which is proportional to temperature dependent London penetration depth $\lambda(T)$ for $d \ll \lambda$[40,41]. From the critical current $I_c(T)$, at which the measured voltage jumps due to the superconducting-to-normal transition, we obtained the corresponding $J_{c,sf}(T)$ and $\lambda(T)$ curves (Fig. 4c and Supplementary Fig. 10), which can be nicely reproduced by the fit based on BCS theory using $\Delta_{SC} \approx 0.38$ meV. The superconducting gap ($\Delta_{SC}$), taken from IrTe$_2$ nanoflakes with different thicknesses, varies linearly with their $T_c$ with a superconducting gap ratio of $2\Delta_{SC}/k_B T_c \sim 5.3$, which is much larger than the BCS

value of 3.53 and $2\Delta_{SC}/k_B T_c \sim 3.7$ of doped bulk IrTe$_2$[37,42] (Fig. 4d). Thus, superconductivity in IrTe$_2$ nanoflakes is in the strong coupling regime, whereas that of doped bulk IrTe$_2$ is in the weak coupling regime.

**Discussion**

Our findings unequivocally emphasise that the superconductivity in IrTe$_2$ nanoflakes, which emerges from the preexisting stripe order, is highly distinct from that in doped bulk IrTe$_2$. On pristine IrTe$_2$ bulk or surface, the $5a_0$ stripe phase undergoes multiple transitions to other nearly-degenerate stripe phases that have different periods such as $8a_0$ and $6a_0$, and also a honeycomb phase[35,43–45]. The complex stripe ordering formation is a result of subtle balance between local interactions of Ir-Ir dimerization and Te-Te depolymerisation. These incipient instabilities and the resulting strong electron-lattice coupling of the parent $5a_0$ stripe phase can facilitate pairing interaction for superconductivity in a proper condition and thereby enhancing the superconducting coupling strength as observed in IrTe$_2$ nanoflakes. This coexisting phase of stripe and superconducting orders, however, cannot be accessed by chemical doping, e.g. Pt doping at the Ir sites. A few % of doping directly perturbs Ir dimerization and melts the stripe order to a quasi-periodic hexagonal order[18,22]. In this case, the onset of superconductivity coincides with disorder-induced melting of the parent order[23], reminiscent of other TMDCs such as Cu-doped TiSe$_2$[3–5].

In contrast, the thickness control of IrTe$_2$ tunes the stripe order without introducing quenched disorders. The stripe order in IrTe$_2$ may be mildly suppressed by the thinning-induced out-of-plane elongation[13,46] or the substrate-induced in-plane tensile strain as opposite to the pressure effect enhancing $T_s$[24]. Typically, the thinning-induced out-of-plane elongation of $\Delta c/c \sim 0.1\%$, as found in TaS$_2$[13,46] and the substrate-induced in-plane strain of $\Delta a/a \sim 0.1$–$0.3\%$ (Supplementary Note 6) are expected in the thinned IrTe$_2$, where $a$ and $c$ are the in-plane and out-of-plane lattice constants, respectively. A recent study on a strained IrTe$_2$ single crystals[47] revealed that only $\sim 0.1\%$ of tensile strain induces the transition between the stripe-charge-ordered phases with different periods of $5a_0$ and $6a_0$, which significantly modifies the electronic structures. Our first principle calculations for electron-phonon coupling constant $\lambda_{ep}$ of the $5a_0$ stripe phase show that the in-plane tensile strain is more effective to drive the system to the structural instability and to enhance $\lambda_{ep}$ than the out-of-plane elongation, which increases $T_c$ by a factor of $\sim 3$ (Supplementary Table 1). While the corresponding critical strain is much larger in calculations and also the calculated $T_c$ remains lower than the measured $T_c \sim 2$ K, these observations imply that the stripe phase of IrTe$_2$ is intrinsically in close proximity to the superconducting phase, which can be accessed by reducing thickness or by thermal quenching[32,33]. It has been known that, in the vicinity of full charge order melting, strong electron-phonon coupling[48,49] can play an important role to promote the superconductivity, as found in the charge-ordered organic metals[50,51]. This appears to be consistent with the enhanced superconducting gap ratio, found in IrTe$_2$ flakes (Fig. 4). Our results unveil the collaborating relationship, rather than the competing one, between the parent stripe and the superconducting orders in IrTe$_2$, highlighting IrTe$_2$ as a unique example among superconducting TMDCs. Further investigations, for examples, scanning tunnelling microscopy and spectroscopy as well as Raman spectroscopy below the $T_c$ are highly desirable to identify the local superconducting gap structure along the charge modulation patterns and the electron-phonon coupling, which would elucidate the interplay of the stripe-charge-order and superconductivity in IrTe$_2$.

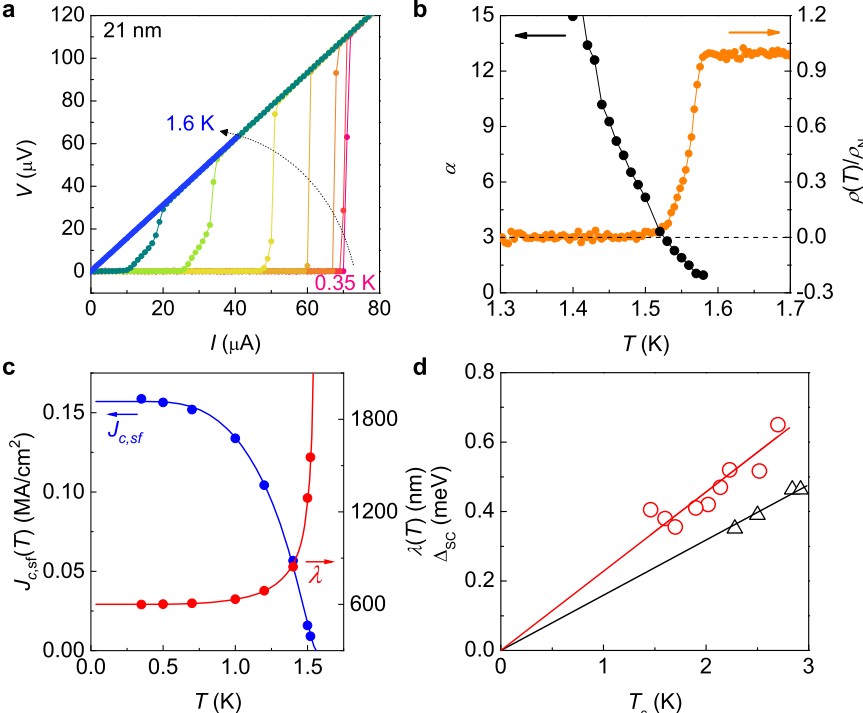

**Fig. 4 Strong superconducting coupling in IrTe$_2$ nanoflakes. a** Current-voltage (*IV*) characteristics at various temperatures for a representative IrTe$_2$ nanoflake with $d = 21$ nm. **b** Temperature dependence of the normalised resistivity and the exponent $\alpha$ for a 21-nm-thick IrTe$_2$ nanoflake. Exponent $\alpha$ is determined from the power-law behaviour $V \propto I^\alpha$ in the *IV* curves, as expected by BKT transition. **c** Critical current density (blue) and the penetration depth (red) as a function of temperature for a 21-nm-thick IrTe$_2$ nanoflake. Solid lines are the fits to the self-critical-current model described in the text. **d** Superconducting gap $\Delta_{SC}$ as a function of $T_c$, extracted from the critical current density for IrTe$_2$ nanoflakes with different $d$ (red). The slope of their linear dependence, corresponding to the superconducting gap ratio, $2\Delta_{SC}/k_B T_c = 5.3$, is much larger than the case of doped bulk IrTe$_2$ (black) from refs. [37,42]. This difference confirms the strong coupling nature of the superconductivity in IrTe$_2$ nanoflakes.

## Methods

**Single crystal growth and bulk properties.** IrTe$_2$ single crystals were synthesised using the Te-flux method[31]. Ir and Te powders were mixed in a stoichiometric ratio Ir:Te = 1:4, heated to 1160 °C for 1 day as sealed in a quartz ampoule, and then cooled. The crystallinity and stoichiometry were confirmed by X-ray diffraction and energy-dispersive X-ray spectroscopy.

**Exfoliation and fabrication of nanoflakes.** We used mechanical exfoliation of bulk single crystals to obtain thin nanoflakes of IrTe$_2$ on top of a Si/SiO$_2$ substrate that had been pre-cleaned in acetone, 2-propanol, and deionised water, then treated by oxygen plasma (O$_2$ = 10 sccm, $P \sim 100$ mTorr) for 5 min. All cleaving and handling were done in inert atmosphere (H$_2$O < 0.1 ppm, O$_2$ < 0.1 ppm) of pure Ar gas except the atomic force microscopy (AFM) measurements. In some cases, a thin h-BN crystal was subsequently transferred onto the IrTe$_2$ nanoflake in Ar atmosphere. We found that the optical contrast, the AFM thickness, and Raman spectra were unchanged even after 1 week in ambient conditions (Supplementary Fig. 1), indicating that the nanoflakes are stable in ambient conditions. To fabricate devices for electrical measurements, we used conventional e-beam lithography to pattern electrodes on top of IrTe$_2$ nanoflakes with metal deposition of Cr(10 nm)/Au(50 nm). The optical microscope image for the typical device is shown in Fig. 1f.

**Transport property measurements.** Transport measurements were performed in a cryogenic ³He refrigerator equipped with a superconducting vector magnet (9/2/2 T). Each measurement wire was filtered by a room-temperature $\pi$ filter and low-temperature $\pi$ and low-pass RC filters at 0.35 K to minimise the electrical noise on the sample. Electrical resistance was measured in standard four-probe configuration using DC delta mode with bias current 1 $\mu$A to 10 $\mu$A determined by the sample resistance and signal to noise ratio. Magnetic fields with desired field orientations were applied by the vector magnet at one cooling without altering sample position.

**Raman spectroscopy.** Raman spectra were obtained using a confocal microscopy set-up with laser beam size of ~ 1 $\mu$m and laser power of ~ 1 mW. A

HeNe laser (632.8 nm) was used to excite IrTe$_2$ flakes in an optical cryostat at normal incidence. The Raman signal was collected in the backscattering configuration and analysed using a monochromator equipped with a liquid nitrogen-cooled silicon CCD. Two linear polarizers in the parallel configuration were placed immediately after the laser and before the monochromator to define the polarization of incident and scattered light, respectively. The crystal orientation relative to the polarisation of the incident light was controlled using a half waveplate between a beam splitter and IrTe$_2$ flakes. The sample position was precisely controlled using a piezo stage.

**Scanning tunnelling microscopy and spectroscopy.** For scanning tunnelling microscopy (STM) and spectroscopy (STS) measurements, IrTe$_2$ nanoflakes were exfoliated in a glove box (H$_2$O < 0.1 ppm, O$_2$ < 0.1 ppm) filled with Ar gas, and transferred onto graphene, grown epitaxially on a 4H-SiC(0001) substrate. The samples were then transferred to a ultrahigh vacuum chamber ($P \leq 1 \times 10^{-10}$ Torr) for STM/STS measurements without any exposure to air to ensure clean surfaces of IrTe$_2$ nanoflakes. STM images were typically obtained using a bias voltage $V_b = -2.5$ V and a tunnelling current $I_t = 20$ pA for large-scale imaging; $V_b = 15$ mV and $I_t = 1$ nA for charge-ordered stripe phases; $V_b = 5$ mV and $I_t = 2$ nA for atomically-resolved images. For STS, we used the lock-in technique with a bias modulation of 7 mV$_{rms}$.

**Density functional theory calculations.** Density functional theory (DFT) calculations for electronic structures and the electron-phonon coupling (EPC) were performed by the Quantum Espresso package implementing the pseudo-potential band method and the density functional perturbation theory[52,53]. We utilised Perdew-Burke-Ernzerhof sol (PBEsol, revised PBE for solid)[54] as an exchange-correlation functional and included the spin-orbit coupling (SOC). The dynamical matrices were calculated using $2 \times 2 \times 2$ $q$-mesh and $16 \times 10 \times 4$ $k$-mesh with 40 Ry energy cutoff. We applied the various in-plane tensile strains in the range of $\Delta a/a \sim 2.1$–3.1% and the fixed compressive strain of $\Delta b/b \sim 0.65$% to model the experimental situations (Supplementary Note 6). Atomic positions were optimised in each case.

## Data availability

The data that support the findings of this study are available from the corresponding authors on request.

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

## Acknowledgements

The authors thank K.T. Ko, G.Y. Jo, Y.K. Bang for fruitful discussion. We also thank H. G. Kim in Pohang Accelerator Laboratory (PAL) for the technical support. This work was supported by the Institute for Basic Science (IBS) through the Center for Artificial Low Dimensional Electronic Systems (no. IBS-R014-D1), the National Research Foundation of Korea (NRF) through SRC (Grant No. NRF-2018R1A5A6075964), and the Max Planck-POSTECH Center for Complex Phase Materials (Grant No. NRF-2016K1A4A4A01922028). J.K., M.J.K. acknowledge the support from the NRF of Korea grant (Grant No. NRF-2017R1C1B2012729 and NRF-2020R1A4A1018935). K.W. and T.T. acknowledge support from the Elemental Strategy Initiative conducted by the MEXT, Japan, Grant Number JPMXP0112101001, JSPS KAKENHI Grant Numbers JP20H00354 and the CREST (JPMJCR15F3), JST. E. F. T. thanks financial support provided by the state assignment of Minobrnauki of Russia (theme Pressure No. AAAA-A18-118020190104-3) and by Act 211 Government of the Russian Federation, contract no. 02.A03.21.0006. S.K. acknowledges the support from the NRF of Korea grant (Grant No. NRF-2019R1F1A1052026) and KISTI supercomputing center (Grant No. KSC-2019-CRE-0172). K.K. acknowledges the support from the NRF of Korea grant (Grant No. 2016R1D1A1B02008461) and Internal R&D programme at KAERI (Grant No. 5244460-21). SWC was partially supported by the NSF under Grant No. DMR-1629059.

## Author contributions

S.P., S.Y.K., and J.S.K. conceived the experiments. S.Y.K. and S.P. fabricated the devices. S.P., S.Y.K., G.S.C., and E.F.T. performed transport property measurements and data analysis. H.K.K., H.W.Y., and T.-H.K. performed scanning tunnelling microscopy/ spectroscopy and data analysis. M.J.K., S.Y.K., T. K., H.K., B.J.K., and J.K. performed Raman spectroscopy and data analysis. S.K. and K.K. performed density functional theory calculation and analysis. C.J.W. and S.-W.C. synthesised the bulk crystals. K.W. and T.T. provided boron nitride crystals. S.P., S.Y.K., J.K., T.-H.K., and J.S.K. co-wrote the manuscript. All authors discussed the results and commented on the paper.

## Competing interests

The authors declare no competing interests.
