## [Peer Review File · Nature Communications]

REVIEWER COMMENTS

Reviewer #1 (Remarks to the Author):

The authors reported the superconductivity emerging from a stripe charge order (SCO) in IrTe₂ nanoflakes. Through mechanical exfoliation, Raman spectroscopy, scanning tunneling microscopy and transport property measurements were carried out on the thinned flakes down to 12 nm. A dome-like superconducting phase was observed as a function of the thickness (d). The superconducting dome is encompassed by the parent stripe phase rather than the common case. The authors attribute the 2D superconductivity of relative thick IrTe₂ flakes as a consequence of coexistence of SCO and superconductivity. This is a good work trying to understand the origin of the superconductivity in thinned IrTe₂ nanoflakes. My comments are listed in the following:

1. The coexistence of SCO and superconductivity: The authors concluded that the SCO coexists with superconductivity from the 2D superconductivity of relative thick flakes and the drastically increased $\xi_c(0)$. It is better that the authors can provide direct evidence the SCO exists below T_c .
2. What is the difference between the result of M. Yoshida et al. (Nano Lett. 2018, 18, 5, 3113–3117) and this work? Why the superconducting state in this work is stable while the state in the work of M. Yoshida et al. is metastable at similar thickness?
3. Page 4. How do the authors define the T_c ? Is it the onset of the transition or the temperature when the resistivity reaches zero?
4. Fig. 1h. What is the error bar of each data points?
5. Why are there multi-steps in Fig. S6c,d,e,f,g,h? Does it mean the multi-phases in the studied samples? For sample with $d = 140$ nm, why the resistivity close to 0.35 K at the low magnetic field is not zero and noisy? Why the data in Fig. S6e go up when approaching zero field?
6. Effect of strain. Though the calculation shows the tensile strain "may play an important role for suppression of the stripe order and appearance of superconductivity in IrTe₂ nanoflakes". It is not convincing to me that the presumed thermal contraction would affect the nanoflakes with the setup proposed by the authors. Is there any existing example that the sample anchored on a substrate by electrodes would experience any significant strain during cooling down? I cannot get the 3% tensile strain taking the expansion coefficient of Si and the temperature difference of 280 K. Would the authors please explain how did they come up with the strain number? The author considered the thermal expansion of the substrate (Si). How about the electrodes and flakes? What is the expansion coefficient of them?
7. What drives the superconductivity in IrTe₂?

Reviewer #2 (Remarks to the Author):

In many superconductors, the superconducting phase appears when other competing ordered phases are suppressed. Superconductivity in chemically doped IrTe₂ also occurs by suppressing the stripe charge ordering phase of the parent material. However, chemical doping introduces a disorder effect into the system. The authors focused on the occurrence of superconductivity without chemical doping when reducing the thickness of IrTe₂, and investigated the relationship between superconductivity and stripe charge ordering phase without the disorder effect.

The authors report a thickness-dependent phase diagram of IrTe₂ based on the results of Raman spectroscopy, scanning tunneling microscopy/spectroscopy, and transport properties experiments. They found that the superconducting phase of IrTe₂ exists in the stripe charge ordering phase,

suggesting that the instability inherent in the parent stripe phase is sufficient to induce superconductivity in IrTe₂ without its complete or partial melting.

The content of this paper is very interesting and worth reading. However, there are several points in this manuscript that need to be revised. I would like to ask the authors to address the following points

[1] I would like the authors to add Raman data for the samples with $d < 30$ nm. The authors should present evidence that the samples with $d < 30$ nm form the stripe charge order to some depth, not only on the surface suggested by STM.

It is of concern that the phase transition temperature of the stripe charge ordering phase is not plotted in the $d < 30$ nm region in Fig. 1h, but the superconducting transition temperature is present. As it appears to contradict the claim, a correction or explanation should be made or added.

[2] For the data of the $d = 140$ nm sample in Fig. 3c, I cannot judge whether the fitting is correct or not because the data points are scattered. The data is important to discuss the relationship between the thickness of the sample and the dimensionality of the superconductivity. Therefore, it is recommended to re-measure.

[3] The authors claim that the out-of-plane coherence length must be longer than the thickness of the samples because the angular dependence of the upper critical field follows that of two-dimensional superconductivity. By the way, it would be better to discuss using actual values of out-of-plane coherence length. Is it possible?

[4] The authors seem to understand that Te-Te polymerization occurs in the stripe charge ordering phase, but I think that the opposite is true. That is, the depolymerization occurs in the stripe charge ordering phase [Y. S. Oh, PRL 110, 127209 (2013)].

[5] In the manuscript, the authors claim to have observed superconductivity at $T_c = 1.43$ - 2.64 K, but it appears to be $T_c < 2.4$ K in Fig. 1g.

[6] I would like the authors to describe the measurement conditions, such as sample thickness and temperature, for each measurement in the figure or figure caption. It makes it easier to read.

[7] What value is given for the sample thickness used in the STM/STS experiments, the value before or after cleaving?

[8] For the future perspective, I would like the authors to discuss or comment on what will be revealed by performing Raman scattering and STM/STS experiments for the IrTe₂ nanoflakes below the superconducting transition temperature.

Reviewer #3 (Remarks to the Author):

In this manuscript, the relationship between superconductivity and a stripe charge order in IrTe₂ nanoflakes is well discussed based on Raman spectroscopy, scanning tunnelling microscopy, and transport property measurements. This material is highly topical, and the coexistence of superconducting phase and charge stripe order is one of the interesting topics in the research region of condensed matter physics. In my opinion, the results reported here are of a high standard and standard approaches to data analysis have been used that lead to robust results. So most part of this manuscript may be worth being published in Nature communications.

However, I feel that there is a lack of the experimental data or explanation about the phase diagram shown in Fig. 1h. In Page 3, the author argued that "we found that the parent stripe phase encompasses the whole superconducting dome in the thickness-dependent phase diagram". Experimental evidence for that is important. Compared with bulk IrTe₂, coexisting region in phase diagram is relatively wide, and physical properties are significantly different as discussed in the manuscript. However, to clarify whether "the stripe phase encompasses the whole superconducting dome" or not, stripe order should be detected in very thin nanoflakes with ~10 nm thickness, where superconductivity cannot be detected as shown in Fig. 1h. I recommend to add the experimental data or some discussion about nanoflakes with 4-21 nm thickness.

I also have some minor comments as shown below. It can be considered as minor corrections.

1. Page 3, line 17 : Authors referred Ref 18-21 as the reference for superconductivity of IrTe₂. I recommend to add other references of Cu_xIrTe₂ and Rh-doped IrTe₂.
M. Kamitani et al., Phys. Rev. B 87, 180501R (2013).
K. Kudo et al., J. Phys. Soc. Jpn. 82, 085001 (2013).
2. Fig. 1a and Fig. 3a : Tick marks in horizontal axis cannot be read by experimental data points.
3. Page 7, line 2 and line 18: Definition of "dSC" is explained twice.
4. Page 7, line 8 and Fig. 3c : Author argue that "For $d = 140$ nm, $Bc2(\theta)$ deviates from the 2D model and approaches the anisotropic 3D model (Fig. 3c)." . Maybe that is true. However, the differences between data and 2D model fitting are small. Furthermore, two data points around -2 degree are not fitted by both 2D and 3D model. Some explanation or modification may be recommended.
5. Page 9, "Methods", "Exfoliation and fabrication of nanoflakes.": There is no argument about the preparation of bulk single crystals of IrTe₂ to obtain nanoflakes. Preparation of bulk IrTe₂ should be shown or some reference should be referred.

Reply to the Reviewers

Answers to reviewer #1's questions

We sincerely appreciate the reviewer's helpful comments and suggestions. We did our best to answer to each single question and we hope our answers satisfy the reviewer.

Q1-1. *The coexistence of SCO and superconductivity: The authors concluded that the SCO coexists with superconductivity from the 2D superconductivity of relative thick flakes and the drastically increased $\zeta c(0)$. It is better that the authors can provide direct evidence the SCO exists below T_c .*

A1-1. We agree with the reviewer that simultaneous measurements of transport, Raman spectroscopy or scanning tunnelling microscopy (STM) measurements on a IrTe₂ flake below T_c would provide direct evidence for the coexistence of the stripe charge order and superconductivity. However, due to the relatively low superconducting transition temperature $T_c \sim 2$ K, it is quite challenging for us to conduct Raman and STM experiments below T_c .

Fig. S3. Raman spectra of IrTe₂ nanoflakes at 4 K. **a**, Raman spectra of nanoflakes with $10 \leq d \leq 174$ nm and bulk at 4 K. **b**, Thickness-dependent phase diagram with a superconducting dome (Black). Red crosses indicate the thickness and temperature points at which the stripe charge-order formation is confirmed by Raman spectroscopy. **c**, Optical images of nanoflakes used for Raman spectroscopy (scale bar, 3 μm). The positions where the laser was focused are indicated by red circles.

Instead, we measured the Raman spectroscopy on IrTe₂ flakes cooled down to 4 K, just above $T_c \sim 2$ K. For fifteen IrTe₂ flakes with different thicknesses, which cover the whole superconducting dome in the thickness-dependent phase diagram, we observed clear evidence of the stripe charge order as shown in Supplementary Fig. S3 in the revised supplementary information. It is highly unlikely that the stripe charge order, well stabilized at 4 K, suddenly disappears before the superconducting transition at $T_c \sim 2$ K. These additional results, together with the drastic increase of the c -axis superconducting coherence length (Fig. 3), strongly support that the stripe-charge-order serves as the parent state for the superconductivity in IrTe₂ flakes.

In the revised manuscript, we include the results of these additional experiments in the supplementary information.

Q1-2. *What is the difference between the result of M. Yoshida et al. (Nano Lett. 2018, 18, 5, 3113–3117) and this work? Why the superconducting state in this work is stable while the state in the work of M. Yoshida et al. is metastable at similar thickness?*

A1-2. We appreciate the reviewer for this valuable comment. In M. Yoshida et. al.'s study, they focused on the metastable superconductivity of IrTe₂ and observed the different behaviours of the superconducting transitions during different cooling procedures. Percolative formation of macroscopic charge-disordered domains, hosting superconductivity, was conjectured as the origin for the observed metastable superconductivity, in the background of stripe-charge-ordered phase. Recent experimental studies [H. Kim et al. Nano Lett. 16, 4260–4265 (2016) and H. Oike et al., Sci. Adv. 4, eaau3489 (2018)] revealed that rapid cooling induces inhomogeneous domain formation of the stripe-charge-ordered and charge-disordered phases in IrTe₂ flakes. The key difference in our work is that we used a much slower cooling rate ~ 0.5 K/min than those in M. Yoshida et. al.'s work. In our experiments, we focused on the slowly-cooled crystals with the homogeneous and reproducible charge-ordered phase and investigated the properties of coexisting superconducting phase.

In the revised manuscript, we clearly pointed out the different cooling rate in our experiments and its effect on the charge-ordering behaviours.

Q1-3. *Page 4. How do the authors define the T_c ? Is it the onset of the transition or the temperature when the resistivity reaches zero?*

A1-3. In the previous version, we took the onset of the resistive transition as the superconducting transition temperature (T_c). In the revised manuscript, we determine T_c with 50 % of the resistive transition and plot them in Fig. 1h with the error bars estimated from the resistive transition width.

Q1-4. *Fig. 1h. What is the error bar of each data points?*

A1-4. For the charge ordering (T_s) and the superconducting (T_c) temperatures from the resistivity measurements, the error bars are defined by the width of the corresponding resistive transition. For T_s from Raman spectroscopy measurements, the error bars are defined by the temperature step of 5 K between the measurements.

We clearly mentioned the definition of the error bars in the Figure caption (Fig. 1).

Q1-5. *Why are there multi-steps in Fig. S6c,d,e,f,g,h? Does it mean the multi-phases in the studied samples?*

A1-5. As the reviewer pointed out, some crystals exhibit multi-steps in magnetoresistance (MR), particularly under the out-of-plane magnetic fields. We found, however, that such multiple steps in MR are suppressed at higher temperatures, close to T_c (Figs. S8 e-h), and under the in-plane magnetic fields (Figs. S8 a-d). These observations indicate that the presence of macroscopic domains with different superconducting properties like T_c or H_{c2} is unlikely to be the origin of the observed multi-steps in MR. Instead, such a behaviour can be induced by the weak links and the associated vortex dynamics in 2D superconductors [M. Morita & S. Okuma, *Phys. C Supercond.* 392–396, 406–409 (2003), N. Paradiso *et al.* *2D Mater.* 6, 025039 (2019) and C. Sharma *et al.*, *Commun. Phys.* 1, 90 (2018)]. As found in the STM results (Fig. 2), charge-ordered domains with different stripe orientations are formed in the single crystalline IrTe₂ flake. The resulting domain boundaries, known to have a high resistance in the stripe-ordered normal state [J. S. You *et al.* *Phys. Rev. B* 103, 045102 (2021)], may serve as weak links between the superconducting domains below T_c . These weak links promote vortex slip motion and are more fragile to external currents or magnetic fields. Therefore, depending on the formation of such domain boundaries, the weak link effect can become strong enough to produce the multi-steps in MR, particularly under the out-of-plane magnetic fields.

In order to explain possible origins of the multi-steps in MR, we added a paragraph in the revised supplementary information.

Q1-6. *For sample with $d = 140$ nm, why the resistivity close to 0.35 K at the low magnetic field is not zero and noisy? Why the data in Fig. S6e go up when approaching zero field?*

A1-6. For the IrTe₂ flakes with $d = 140$ nm, the measured resistance is the smallest among the studied samples due to its large thickness. We used a bias current $I = 5$ μ A for the measurements, which is small enough to avoid the heating effect, but not high enough to obtain clear data over the noise level. The seemingly non-zero resistance in Figs. S8d and S8h is due to the relatively large noise level during the measurements. We note that the zero resistance state is clearly observed in the more careful measurements on the same crystal in Fig. S9d of the revised supplementary information.

Fig. S9. Angle dependent B_{c2} curves determined by different criteria. **a–d**, Magnetic field dependence of the normalized resistivity curves with various angles in IrTe₂ nanoflakes with thickness of 21 (**a**), 56 (**b**), 90 (**c**), and 140 nm (**d**) **e–h**, Corresponding B_{c2} depending on angles, determined by 50% resistive transition for the nanoflakes. The measurements were done at $T=0.35$ K, otherwise noted. Red and blue dashed lines are the fits of the whole angle dependent data to the 2D Tinkham and 3D Ginzburg-Landau models, respectively. **i–l**, Magnified angle dependent B_{c2} near the in-plane magnetic field for $|\theta| \leq 1^\circ$. Red and blue solid lines are the fits of $B_{c2}(\theta)$ data to the 2D Tinkham and 3D Ginzburg-Landau models, respectively. The dashed lines are the fit of the whole angle dependent data as shown in **e–h**.

Q1-7. Effect of strain. Though the calculation shows the tensile strain “may play an important role for suppression of the stripe order and appearance of superconductivity in IrTe₂ nanoflakes”. It is not convincing to me that the presumed thermal contraction would affect the nanoflakes with the setup proposed by the authors. Is there any existing example that the sample anchored on a substrate by electrodes would experience any significant strain during cooling down? I cannot get the 3 % tensile strain taking the expansion coefficient of Si and the temperature difference of 280 K. Would the authors please explain how did they come up with the strain number? The author considered the thermal expansion of the substrate (Si). How about the electrodes and flakes? What is the expansion coefficient of them?

A1-7. We really appreciate the reviewer’s critical and helpful comments. In the previous

manuscript, we estimated the tensile strain of $\sim 3\%$ from comparison of thermal expansion coefficients of Si ($\alpha \sim +2.5 \times 10^{-6} \text{ K}^{-1}$) and IrTe₂ ($\alpha \sim +110 \times 10^{-6} \text{ K}^{-1}$ along the a -axis in the stripe-charge-ordered phase) [T. Toriyama et al., JPSJ, 83, 033701 (2014)]. As the reviewer pointed out, the substrate-induced strain, estimated in other 2D materials e.g. MoS₂ ($\alpha \sim +8 \times 10^{-6} \text{ K}^{-1}$) [Y. Ding et al., RSC Adv., 5, 18391 (2015)], is typically 0.1 - 0.3 % [G. Plechinger et al., 2D Mater., 2, 015006 (2015)]. The large thermal contraction of IrTe₂, distinct from that of Si, is due to the stripe charge ordering. However, we noticed that in the stripe charge ordered phase, the expansion coefficient along the b -axis is negative, $\alpha \sim -21 \times 10^{-6} \text{ K}^{-1}$, opposite to that of the a -axis. Therefore, in the real samples, one can expect that the substrate-induced strain would be released by the domain formation with different stripe orientations. As shown in Fig. 2 of the main text as well as Figs. S5 and S6 in the supplementary information indeed show such domain structures in IrTe₂ flakes. In this regard, we conclude that the actual strain applied to IrTe₂ flakes would be much more moderate than our initial estimate of $\sim 3\%$.

We would like to point out that even with a moderate strain, the charge-ordered ground state and also the appearance of the superconductivity can be significantly changed. A recent study on a strained IrTe₂ single crystals [C. W. Nicholson et al. Commun. Mater. 2, 25 (2021)] has reported that only $\sim 0.1\%$ of tensile strain strongly suppresses the known stripe phase with a periodicity $5a_0$ (a_0 is the in-plane unit cell parameter), but stabilizes a hidden phase with a period of $6a_0$. Among the stripe ordered phases in IrTe₂, these $5a_0$ and $6a_0$ phases have the lowest and the highest Ir-Ir dimer densities, respectively. Therefore, this strain-induced changes significantly modify the charge transfer between Ir and Te states and thus the electronic structures. Such an extreme sensitivity of the ordered phases to the strain cannot be fully captured in the density-functional-theory (DFT) based calculations. We found that in calculations the $5a_0$ phase remains stable up to $\sim 3\%$ of tensile strain, in contrast to the experiments, while the systematic phonon softening related to the structural instability and the resulting enhancement of the electron-phonon coupling are clearly observed upon increasing the strain. Therefore, the experimentally observed extreme sensitivity of the $5a_0$ stripe phases strongly suggests that the $5a_0$ stripe phase is much closer to the structural instability than expected in DFT-based calculations.

In the revised manuscript, we revised our statement about the estimate of the substrate-induced strain and clearly mentioned that the substrate-induced strain on the IrTe₂ flakes would be moderate. Possible role of moderate strain due to the extreme sensitivity of the stripe-charge-ordered phase to the strain is discussed in the revised main text and also the supplementary information.

Q1-8. *What drives the superconductivity in IrTe₂?*

A1-8. In our experimental study, the underlying mechanism of the superconductivity in IrTe₂ remains to be clarified. As explained in A1-7, we conjectured that in IrTe₂ flakes the thinning-induced out-of-plane elongation or the substrate-induced in-plane strain can significantly affect the stability of the $5a_0$ stripe phases. It has been known that, in the vicinity of full charge order melting, strong electron-phonon coupling [J. Merino and R. H. McKenzie, Phys. Rev. Lett. 87, 237002 (2001), A. Foussats et al. Phys. Rev. B 72, 020504(R) (2005)] can play an important

role to promote the superconductivity, as found in the charge-ordered organic metals e.g. β -(BEDT-TTF)₂SF₅CH₂CF₂SO₃ [S. Kaiser, et al. Phys. Rev. Lett. 105, 206402 (2010), A. Girlando et al. Phys. Rev. B 89, 174503 (2014)]. This appears to be consistent with the enhanced superconducting gap ratio, found in IrTe₂ flakes (Fig. 4), but not in bulk Pt-doped IrTe₂ where the superconductivity appears after full suppression of the charge order [K. Takubo et al., Phys. Rev. B. 97, 205142 (2018)]. Thus our findings demonstrate that transition-metal-dichalcogenides can host the novel phase with coexisting charge order and superconductivity, like organic charge-transfer crystals, which will stimulate further experimental and theoretical studies on the interplay between charge order and superconductivity.

In the revised manuscript, we added a few sentences to discuss more on the interplay of charge order and superconductivity with a possible connection to the case of charge-ordered organic metals.

We sincerely appreciate the reviewer's helpful comments and suggestions. We did our best to answer to each single question and we hope our answers satisfy the reviewer.

Q2-1. *I would like the authors to add Raman data for the samples with $d < 30$ nm. The authors should present evidence that the samples with $d < 30$ nm form the stripe charge order to some depth, not only on the surface suggested by STM. It is of concern that the phase transition temperature of the stripe charge ordering phase is not plotted in the $d < 30$ nm region in Fig. 1h, but the superconducting transition temperature is present. As it appears to contradict the claim, a correction or explanation should be made or added.*

A2-1. Following the reviewer 2's suggestion, we conducted additional Raman and STM measurements on IrTe₂ flakes in a range less than 30 nm and also thicker samples as shown in Figs. S3 and S6 of the revised supplementary information. Although we were not able to determine the critical temperature of the stripe-charge ordering, we obtained the experimental evidence of the stripe-charge-order formation, using the Raman spectroscopy, for fifteen nanoflakes with different thicknesses, ranging from 10 nm to 174 nm (Fig. S3c), which cover

Fig. S3. Raman spectra of IrTe₂ nanoflakes at 4 K. **a**, Raman spectra of nanoflakes with $10 \leq d \leq 174$ nm and bulk at 4 K. **b**, Thickness-dependent phase diagram with a superconducting dome (Black). Red crosses indicate the thickness and temperature points at which the stripe charge-order formation is confirmed by Raman spectroscopy. **c**, Optical images of nanoflakes used for Raman spectroscopy (scale bar, 3 μm). The positions where the laser was focused are indicated by red circles.

the whole superconducting dome in the thickness-dependent phase diagram. For all the flakes cooled down to 4 K, just above $T_c \sim 2$ K (Fig. S3a), we observed multiple peak splitting of Raman modes without any signature of the Raman modes for the high-temperature normal phase. It is highly unlikely that the stripe charge order, well stabilized at 4 K, suddenly disappears before the superconducting transition at $T_c \sim 2$ K. These additional results, together with the drastic increase of the c -axis superconducting coherence length (Fig. 3), strongly support that the stripe-charge-order serves as the parent state for the superconductivity in IrTe₂ flakes.

Q2-2. For the data of the $d = 140$ nm sample in Fig. 3c, I cannot judge whether the fitting is correct or not because the data points are scattered. The data is important to discuss the relationship between the thickness of the sample and the dimensionality of the superconductivity. Therefore, it is recommended to re-measure.

A2-2. Following the reviewer 2's suggestion, we conducted additional angle dependent $B_{c2}(\theta)$ measurements on the 140 nm-thick sample at a lower temperature $T = 0.35$ K than the previous measurements ($T = 1.6$ K). Since the change in $B_{c2}(\theta)$ within $|\theta| \leq 1^\circ$ is significantly reduced to only a few % in this case, which contrasts to the case of the thinner samples, *e.g.* $\sim 26\%$ for the 21 nm-thick sample, it was challenging for us to obtain more precise results of $B_{c2}(\theta)$ with a better resolution. However, we found that the experimental data show a clear difference from the anisotropic 3D Ginzburg-Landau model, but agree reasonably well with the 2D Tinkham model, as shown in Figure below. In addition, we measured $B_{c2}(\theta)$ for the 120 nm-thick sample, $T = 0.35$ K, which shows the similar behaviours with the 140 nm-thick sample. These additional results strongly suggest that the 2D superconductivity is realized even for relatively thick IrTe₂ flakes.

In the revised manuscript, we clearly state this point and revised the sentence explaining the results in the main text. The new data are presented in Fig. 3c and Fig. S9l in the revised manuscript.

Fig. R1. Angle dependence of B_{c2} near the in-plane magnetic field for for $|\theta| \leq 1^\circ$ with $d = 140$ and 120 nm at $T = 0.35$ K. The $B_{c2}(\theta)/B_{c2}(0)$ follows reasonably well the 2D Tinkham model (red), rather than the 3D Ginzburg-Landau model (blue).

Fig. 3. Two-dimensional superconductivity of IrTe₂ nanoflakes. **a**, Magnetic field dependence of $\rho(H)$ of a 56-nm-thick IrTe₂ nanoflake, measured with different field orientations θ at $T = 0.35$ K. **b**, Upper critical field B_{c2} as a function of field angle θ for IrTe₂ nanoflakes with different thickness (d) at $T = 0.35$ K, together with the fit (solid line) to the 2D Tinkham model. Inset: the anisotropy factor $\Gamma = \mathbf{B}_{c2}^{ab}/\mathbf{B}_{c2}^c$ as a function of d , following $1/d$ dependence (greyline). Schematic illustration shows the field orientation θ . **c**, Angle dependence of $B_{c2}(\theta)$ of IrTe₂ nanoflakes with $d = 56$ and 140 nm at $T = 0.35$ K. Good agreement with the 2D Tinkham model (red), rather than the 3D Ginzburg-Landau model (black), confirms the 2D superconductivity. **d**, Normalised $B_{c2}/B_{c2}(0)$ as a function of T/T_c for IrTe₂ nanoflakes with different d . All data collapse into dashed lines described by $1-T/T_c$ and $(1-T/T_c)^{0.5}$ for $B\parallel c$ (open circles) and $B\parallel ab$ (solid circles), respectively. **e**, Ginzburg-Landau coherence length ξ_{ab} (square) and the effective superconducting thickness d_{sc} (circle) as a function of d . ξ_{ab} is nearly independent of d , whereas d_{sc} grows linearly with d ($d_{sc} \sim 0.8d$) and exceeds ξ_c of doped bulk IrTe₂. **f**, Schematic illustration of the size effect of vdW superconductors. In normal vdW superconductors with a large anisotropy $\xi_c \ll \xi_{ab}$, 2D superconductivity appears only for a-few-layer-thick crystals. In IrTe₂ with a stripe order and the resulting cross-layer quasi-2D state, the increased $\xi_c \sim \xi_{ab}$ induces 2D superconductivity in relatively thick nanoflakes.

Q2-3. *The authors claim that the out-of-plane coherence length must be longer than the thickness of the samples because the angular dependence of the upper critical field follows that of two-dimensional superconductivity. By the way, it would be better to discuss using actual values of out-of-plane coherence length. Is it possible?*

A2-3. In our experiments, all the flakes with thickness up to 140 nm show two-dimensional (2D) superconductivity. In this case, the upper critical field for the in-plane magnetic field ($H \parallel ab$) is determined by the thickness of the superconducting layer, not by the out-of-plane coherence length. From the experiments, we directly estimate the thickness of superconducting layers, which should be smaller than the out-of-plane coherence length, and provide the lower bound of the out-of-plane coherence length. For the 140 nm-thick sample showing the 2D superconductivity, the estimated superconducting layer is ~ 100 nm, which serves as a lower bound of the out-of-plane coherence length.

Q2-4. *The authors seem to understand that Te-Te polymerization occurs in the stripe charge ordering phase, but I think that the opposite is true. That is, the depolymerization occurs in the stripe charge ordering phase [Y. S. Oh, PRL 110, 127209 (2013)].*

A2-4. We appreciate the reviewer's helpful comments. We corrected the sentence as the reviewer 2 suggested.

Q2-5. *In the manuscript, the authors claim to have observed superconductivity at $T_c = 1.43$ - 2.64 K, but it appears to be $T_c < 2.4$ K in Fig. 1g.*

A2-5. As the reviewer 2 suggested, we displayed more data in Fig. 1g for covering the full range of the sample thickness and also the full range of T_c . In the revised Fig. 1g, we presented the temperature dependent resistivity curves corresponding to the samples with a minimum T_c , ($d = 90$ nm), a maximum T_c ($d = 53$ nm) and no superconductivity ($d = 12$ nm).

Fig. 1. Structure and phase diagram of IrTe₂ nanoflakes. **a**, Schematic phase diagrams of TMDs as a function of control parameter p , showing commensurate (C), incommensurate charge order (IC), and superconductivity (SC). Three different types of dome-shaped superconducting phase diagram, where the dome lies at the centre of a presumed quantum

critical point (top), near the C-IC transition line (middle), or well inside the parent order (bottom). **b**, Crystal structure of IrTe₂. **c,d**, Schematic illustrations of the stripe order in IrTe₂ below T_s . The Ir-Ir dimerization (red) with a modulation vector $q = (\frac{1}{5}, 0, \frac{1}{5})$ is depicted (blue shade) on the triangular Ir layer (**c**) and across the stacked layers (**d**). The crystallographic axes for the high- T (a , b , and c) and the low- T (a^* , b^* , and c^*) structures are shown, together with the unit cell of the stripe phase (orange box). **e**, Temperature dependence of the normalised resistivity $\rho(T)/\rho(300\text{ K})$ for IrTe₂ crystals with thickness (d) of 21, 56, and 90 nm. For clarity, $\rho(T)/\rho(300\text{ K})$ curves are offset vertically. Transition temperatures $T_{s,\text{up}}$ and $T_{s,\text{dn}}$ are determined (arrows) in opposite temperature sweeps. **f**, Optical microscope image of a 56-nm-thick IrTe₂ crystal. **g**, $\rho(T)/\rho(3\text{ K})$ curves for IrTe₂ crystals with $d = 12 - 140\text{ nm}$. **h**, Phase diagram of IrTe₂ nanoflakes as a function of thickness d , obtained by transport (circle) and Raman spectroscopy (square) measurements. The transition temperatures $T_{s,\text{up}}$ (red) and $T_{s,\text{dn}}$ (blue) during warming and cooling are plotted with the superconducting transition temperature T_c (black), scaled by a factor of 10 for clarity. The error bars from the resistivity and Raman spectroscopy are defined by the width of the corresponding resistive transitions and the temperature step of 5 K between the measurements, respectively.

Q2-6. I would like the authors to describe the measurement conditions, such as sample thickness and temperature, for each measurement in the figure or figure caption. It makes it easier to read.

A2-6. As the reviewer 2 suggested, we added the measurement conditions in the Figure and figure captions, as shown in e.g. Fig. 2 below.

Fig. 2. Stripe charge ordering formation in IrTe₂ nanoflakes. **a,b**, Raman spectra (**a**) and corresponding temperature dependent Raman frequency (**b**) of a 65-nm-thick IrTe₂ nanoflake at various temperatures during cool-down and warm-up procedures. At $T > T_s$, the Raman modes, E_g at 126 cm^{-1} and A_{1g} at 165 cm^{-1} , split into multiple peaks as the flake forms the

stripe charge order in **a**. The temperature ranges for the normal (red), the stripe (blue), and intermediate coexistence (yellow) phases, are identified in **b**, during cooling (upper panel) and warming (lower panel). Transition temperatures $T_{s, \text{dn}}$ and $T_{s, \text{up}}$ are indicated by the arrows. **c**, Spatial profile of Raman intensity map for 129 cm^{-1} for a 43-nm-thick IrTe₂ nanoflake at $T \sim 70 \text{ K}$. Inset: optical microscope image of the flake. **d**, Large-scale scanning tunnelling microscopy (STM) image of a typical thin IrTe₂ flake at $T = 85 \text{ K}$ (scale bar, 300 nm). The flake with $d = 20 \text{ nm}$ has only stripe-phase charge-ordered domains, illustrated by red, green, and blue lines. Black lines indicate domain boundaries between three equivalent stripe-phase charge-ordered domains. **e**, Atomically resolved STM image at $T = 85 \text{ K}$ representing a uniform striped area with 5×1 surface reconstruction (scale bar, 2 nm). **f**, Zoomed-in STM image of two charge-ordered phases indicated by red square in **d** showing that the two phases coexist at the boundary (scale bar, 20 nm). **g**, Scanning tunnelling spectroscopy (STS) spectra at $T = 85 \text{ K}$ taken on IrTe₂ nanoflakes with $d = 47, 80, \text{ and } 148 \text{ nm}$, as indicated in the plot.

Q2-7. *What value is given for the sample thickness used in the STM/STS experiments, the value before or after cleaving?*

A2-7. The sample thickness of the IrTe₂ nanoflakes is determined after cleaving. As described in Methods, we obtained IrTe₂ nanoflakes on top of a graphitized SiC substrate, whose thickness as well as the topography is measured using scanning tunnelling microscopy.

Q2-8. *For the future perspective, I would like the authors to discuss or comment on what will be revealed by performing Raman scattering and STM/STS experiments for the IrTe₂ nanoflakes below the superconducting transition temperature.*

A2-8. We appreciate the reviewer's valuable suggestion. As explained in **A2-1**, the additional Raman spectroscopy data taken at $T = 4 \text{ K}$, just above $T_c \sim 2 \text{ K}$, suggest that the stripe-charge-ordered phase remains stabilized, below T_c . However, as the reviewer #2 pointed out, by carefully monitoring the Raman spectra across T_c or identifying the local superconducting gap structure along the charge modulation patterns using STM/STS, we could address how the stripe-charge-order affects the electron-phonon coupling or superconducting properties in IrTe₂. As the reviewer suggested, we emphasize that such experiments are highly desirable to elucidate the unusual superconducting phase of IrTe₂ nanoflakes, coexisting with stripe-charge-order, in the revised manuscript.

We sincerely appreciate the reviewer's helpful comments and suggestions. We did our best to answer to each single question and we hope our answers satisfy the reviewer.

Q3-1. *I feel that there is a lack of the experimental data or explanation about the phase diagram shown in Fig. 1h. In Page 3, the author argued that “we found that the parent stripe phase encompasses the whole superconducting dome in the thickness-dependent phase diagram”. Experimental evidence for that is important. Compared with bulk IrTe₂, coexisting region in phase diagram is relatively wide, and physical properties are significantly different as discussed in the manuscript. However, to clarify whether “the stripe phase encompasses the whole superconducting dome” or not, stripe order should be detected in very thin nanoflakes with ~10 nm thickness, where superconductivity cannot be detected as shown in Fig. 1h. I recommend to add the experimental data or some discussion about nanoflakes with 4-21 nm thickness.*

A3-1. We appreciate the reviewer's helpful suggestion. Following the reviewer 3's suggestions, we conducted additional Raman spectroscopy measurements and scanning tunnelling microscopy (STM) measurements on IrTe₂ flakes with the thickness less than 30 nm. As shown

Fig. S3. Raman spectra of IrTe₂ nanoflakes at 4 K. **a**, Raman spectra of nanoflakes with $10 \leq d \leq 174$ nm and bulk at 4 K. **b**, Thickness-dependent phase diagram with a superconducting dome (Black). Red crosses indicate the thickness and temperature points at which the stripe charge-order formation is confirmed by Raman spectroscopy. **c**, Optical images of nanoflakes used for Raman spectroscopy (scale bar, 3 μ m). The positions where the laser was focused are indicated by red circles.

Fig. S6. Stripe charge order formation on IrTe₂ nanoflakes at 85 K. **a–c**, STM topographic images of nanoflakes with thicknesses of **(a)** 11 nm, **(b)** 13.5 nm, and **(c)** 20 nm. All nanoflakes exhibit only the stripe charge ordering without the hexagonal phase. **d–f**, Atom-resolved STM images of **a–c** representing the dominant stripe charge ordering with period of $5a_0$. Note that **c** and **f** are the same as Fig. 2f and 2e of the main text, respectively. Scale bar, **(a–c)** 10 nm and **(d–f)** 2 nm.

in Fig. S3 in the revised supplementary information, the stripe charge order remains stabilized at 4 K, just above the superconducting transition temperature $T_c \sim 2$ K in the IrTe₂ flakes with various thicknesses, from 10 nm to 174 nm, which cover the whole superconducting dome in the thickness-dependent phase diagram. Although we were not able to determine the transition temperatures, these additional results unambiguously confirm that the stripe-charge-order phase is stabilized down to $d \sim 10$ nm and encompasses the whole superconducting dome. We also found that even thinner flakes (11 nm and 13.5 nm) also form only the stripe charge ordering without hexagonal phases by using STM (Fig. S6).

In the revised supplementary information, we added two figures [Fig. S3 and S6] in order to make this point clear.

Q3-2. Page 3, line 17 : Authors referred Ref 18-21 as the reference for superconductivity of IrTe₂. I recommend to add other references of Cu_xIrTe₂ and Rh-doped IrTe₂.

M. Kamitani *et al.*, *Phys. Rev. B* 87, 180501R (2013).

K. Kudo *et al.*, *J. Phys. Soc. Jpn.* 82, 085001 (2013).

A3-2. We appreciate the reviewer's kind comment and recommendation. We added two references in the revised manuscript as suggested.

Q3-3. Fig. 1a and Fig. 3a : Tick marks in horizontal axis cannot be read by experimental data points.

A3-3. As the reviewer #3 suggested, we revised the tick marks to be clearly read.

Q3-4. Page 7, line 2 and line 18: Definition of “dsc” is explained twice.

A3-4. We deleted duplicated sentences.

Q3-5. Page 7, line 8 and Fig. 3c : Author argue that “For $d = 140$ nm, $B_{c2}(\theta)$ deviates from the 2D model and approaches the anisotropic 3D model (Fig. 3c).” . Maybe that is true. However, the differences between data and 2D model fitting are small. Furthermore, two data points around -2 degree are not fitted by both 2D and 3D model. Some explanation or modification may be recommended.

A3-5. In order to clarify this issue, we conducted additional angle dependence measurements on the 140 nm-thick and 120 nm-thick IrTe₂ flakes at $T = 0.35$ K. Since the change in $B_{c2}(\theta)$ for $|\theta| \leq 1^\circ$ is significantly reduced to only a few %, in contrast to the case of the thinner samples, e.g. ~ 26 % for 21 nm-thick sample, it was challenging for us to obtain more precise results with a better resolution of $B_{c2}(\theta)$. However, we found that the experimental data show a clear difference from the anisotropic 3D model in both cases but agree reasonably with the 2D model, as shown in Figure below.

In the revised manuscript, we clearly state this point revised the sentence explaining the results in the main text.

Fig. R1. Angle dependence of B_{c2} near the in-plane magnetic field for for $|\theta| \leq 1^\circ$ with $d = 140$ and 120 nm at $T = 0.35$ K. The $B_{c2}(\theta)/B_{c2}(0)$ follows reasonably well the 2D Tinkham model (red), rather than the 3D Ginzburg-Landau model (blue).

Q3-6. Page 9, “Methods”, “Exfoliation and fabrication of nanoflakes.”: There is no argument about the preparation of bulk single crystals of IrTe₂ to obtain nanoflakes. Preparation of bulk

IrTe₂ should be shown or some reference should be referred.

A3-6. As the reviewer suggested, we included a section for introducing the preparation of bulk single crystals of IrTe₂ in “Method”.

REVIEWERS' COMMENTS

Reviewer #1 (Remarks to the Author):

This is a good work to try to understand the interplay between superconductivity and stripe charge order (SCO) in thinned IrTe₂ flakes. The authors have done a lots of measurements by transport, Raman and STM/S. They have successfully answered all the comments raised by me point-by-point. Although they can not provide the direct evidence for the coexistence of SCO and superconductivity, however, they indeed present new Raman results which show the existence of SCO at 4K, in the vicinity of SC dome. I am convinced with their replys. So I suggest the manuscript can be accepted for publication in Nature Communications.

A few minor comments:

1. in page 6, when discussing the STS measurements, I can not understand the sentences about "The dip feature of the normalised STS spectra, found in bulk IrTe₂, weakens as the thickness of IrTe₂ decreases. This response implies that the observed superconductivity in thin nanofakes may be related to the higher DOS near EF with decreasing thickness." The explanation of dip disappearing, doesn't make any sense for me.
2. in page 7, the authors estimate the out-of-plane coherence length is larger than 100 nm in IrTe₂ nanoflakes, much larger than 25 nm of bulk samples and the in-plane coherence length 70 nm. On the other hand, they claims the 2D superconductivity of IrTe₂ nanoflakes. I feel a little bit confusing, the authors may explain it by adding some words.

Reviewer #2 (Remarks to the Author):

The authors' responses are satisfactory. They have properly revised the manuscript. I recommend the revised manuscript for publication.

Reviewer #3 (Remarks to the Author):

Revised manuscript and replies are well written. I recommend the publication of this manuscript.

Reply to the Reviewers

Answers to reviewer #1's questions

We sincerely appreciate the reviewer's helpful comments and suggestions. We did our best to answer to each single question and we hope our answers satisfy the reviewer.

Q1-1. *In page 6, when discussing the STS measurements, I can not understand the sentences about "The dip feature of the normalised STS spectra, found in bulk IrTe₂, weakens as the thickness of IrTe₂ decreases. This response implies that the observed superconductivity in thin nanoflakes may be related to the higher DOS near EF with decreasing thickness." The explanation of dip disappearing, doesn't make any sense for me.*

A1-1. Following the reviewer's suggestion, we delete the sentence about the dip features of the STS spectra.

Q1-2. *In page 7, the authors estimate the out-of-plane coherence length is larger than 100 nm in IrTe₂ nanoflakes, much larger than 25 nm of bulk samples and the in-plane coherence length 70 nm. On the other hand, they claim the 2D superconductivity of IrTe₂ nanoflakes. I feel a little bit confusing, the authors may explain it by adding some words.*

A1-2. As the reviewer suggested, we added a sentence, clarifying the two-dimensional superconductivity in IrTe₂ nanoflakes, which manifests the distinct nature of superconducting properties of IrTe₂ nanoflakes, as compared to the doped bulk IrTe₂.

List of the changes

In the main text

1. Following the reviewer #1's suggestion, we deleted the sentences in paragraph 8, explaining dip features of STS spectra. (A1-1)
2. Following the reviewer #1's suggestion, we added a sentence explaining for 2D superconductivity of IrTe₂ nanoflakes and their unusual properties. (A1-2)

[paragraph 10, page 7]

“should be larger than the maximum value of $d_{SC} \sim 100$ nm, obtained in experiments (Fig. 3e)... These findings clearly indicate that the superconducting characteristics of IrTe₂ nanoflakes are distinct from those of the doped IrTe₂”